# Action of Carnosic Acid Against Melanoma: A Strategy for Selective Radiosensitization with Protection of Non-Tumoral Cells

**DOI:** 10.3390/cimb47100845

**Published:** 2025-10-14

**Authors:** Amparo Olivares, Isabel de la Fuente, Daniel Gyingiri Achel, Ana María Mercado, José Antonio Garcia-Gamuz, María del Rosario Tudela, Miguel Alcaraz

**Affiliations:** 1Radiology and Physical Medicine Department, School of Medicine, Campus de Excelencia Internacional de Ámbito Regional (CEIR)–Campus Mare Nostrum (CMN), University of Murcia, 30100 Murcia, Spain; amparo.o.r@um.es (A.O.); misabel.de@um.es (I.d.l.F.); anamaria.mercado@um.es (A.M.M.); gamuz@um.es (J.A.G.-G.); 2Department of Medical Physics, School of Nuclear and Allied Sciences, University of Ghana, Atomic Campus, Accra GE-292-9709, Ghana; daniel.achel@gaec.gov.gh; 3Health Sciences Department, UCAM—Universidad Católica de Murcia, Campus de los Jerónimos 135, 30107 Murcia, Spain; mrtudela@ucam.edu

**Keywords:** radiation effects, radioprotectors, radiosensitizers, melanoma, B16F10, PNT2, SK-MEL-1, Melan A

## Abstract

Carnosic acid (CA) is a phenolic diterpene with high antioxidant activity that supports its radioprotective capacity. This study aims to determine whether the radiosensitizing effect of CA established in B16F10 melanoma cells also occurs in other melanin-producing cells. Cell survival analysis, apoptosis, intracellular glutathione levels, and cell cycle progression were evaluated by comparing radiosensitive cells (PNT2) with radioresistant melanin-producing cells (MELAN A, SK-MEL-1, and B16F10). In PNT2 cells, CA exhibited radioprotective capacity, with 100% cell survival after exposure to 20 Gy of X-rays (*p* < 0.001), decreasing apoptosis (*p* < 0.001) and increasing the GSH/GSSG ratio (*p* < 0.01), without significant modification in cell cycle progression. However, CA administration to irradiated cells failed to exert radioprotection in MELAN A and SK-MEL-1 cells, and even doubled cell death in B16F10 cells (*p* < 0.001). Specifically, CA did not alter apoptosis or prevent the decrease in GSH/GSSG ratio in MELAN A and SK-MEL-1 cells, while it intensified radiation-induced cell cycle disruptions in all melanin-producing cells. All of these led to a loss of radioprotective capacity in the melanin-producing cells (MELAN A and SK-MEL-1) and even induced a radiosensitizing effect in B16F10 cells. Understanding the mechanisms of action of substances such as CA could promote new applications that protect healthy cells and exclusively damage neoplastic cells when both are present within the same irradiated volume in cancer patients requiring radiotherapy.

## 1. Introduction

Melanoma is known to have the highest mortality rate among skin cancer patients worldwide [1,2,3]. The current therapeutic approach to advanced human melanoma includes surgery as the first line of treatment in localized stages, while in metastatic phases, immunotherapy, targeted therapies, and in selected cases, adjuvant or palliative radiotherapy are used. However, melanoma is notoriously resistant to chemotherapy and has a limited response to conventional radiotherapy, which warrants the search for novel strategies to improve tumor radiosensitivity [4,5].

In the experimental context, the B16F10 melanoma cell line is a mouse metastatic cell line that is frequently used as a research model of melanomas to assess the response to ionizing radiation and other modulating agents [5,6]. Previous studies have shown that these cells are characterized by high radioresistance attributed to their endogenous antioxidant capacity and the activation of DNA damage repair mechanisms [7,8]. However, it has been described that the administration of certain antioxidants and known radioprotective substances, such as carnosol [9], rosmarinic acid [10,11,12], and carnosic acid (CA) [13,14], paradoxically do not confer radioprotection to B16F10 cells; on the contrary, these compounds can exert a radiosensitizing effect, increasing radiation-induced cell death and raising the enhancement ratio (ER) to values greater than 2.2 [9,12,14]. This paradoxical phenomenon has been linked to melanogenesis activation and intracellular redox system disruption—specifically, glutathione (GSH) depletion. GSH is a key antioxidant that scavenges reactive oxygen species (ROS) induced by ionizing radiation; its depletion impairs endogenous antioxidant defenses, allowing accumulated ROS to cause lethal DNA damage and thus enhancing radiation cytotoxicity [9,10,12]. This paradoxical radiosensitizing effect has also been related to the metabolic adaptations associated with the high metastatic capacity of these cells, to specific characteristics of their enzyme production, to their enormous cellular proliferation and even to the change in microenvironmental pH produced by tumor necrosis associated with their rapid growth [12,14].

In this study we aim to also evaluate the existence of this paradoxical radiosensitizing effect in other normal murine melanocyte cell lines (Melan A) and in human metastatic melanoma cells (SK-MEL-1). The investigation of compounds capable of modulating responses to ionizing radiation—enhancing tumor radiosensitivity while selectively protecting normal cells—represents a promising strategy to optimize therapeutic efficacy in melanoma. This approach opens new avenues for personalized radiotherapy by increasing tumor damage without elevating toxicity in healthy tissues [10,12].

## 2. Materials and Methods

### 2.1. Carnosic Acid

Carnosic acid (CA) (C20H28O4) (4AR,10aS)-5,6-dihydroxy-1,1-dimethyl-7-propan-2-yl-2,3,4,9,10,10a-hexahydrophenanthrene-4a-carboxylic acid) from Rosmarinus officinalis with a purity of ≥91% was purchased from Sigma-Aldrich (Madrid, Spain).

### 2.2. MTT and XTT Cytotoxicity Assay

To investigate the cell survival by means of the MTT assay for adherent cells and the XTT assay for cells in suspension, we used four cell lines with different degrees of resistance to ionizing radiation [15]: human prostate epithelial cells PNT2, a cell line traditionally considered to be radiosensitive and which we used as reference control cells; and three traditionally melanin-producing cell lines considered radioresistant: Melan A (melanocytes of murine origin), SK-MEL-1 (human metastatic melanoma cells) and B16F10 (murine melanoma cells).

PNT2 cells were sourced from the European Collection of Authenticated Cell Cultures (ECACC) (catalog number 95012613, UK) and cultured in RPMI-1640 medium supplemented with 2 mM glutamine. B16F10 cells, kindly provided by Dr. Hearing from the National Cancer Institute (Bethesda, MA, USA), were maintained in a 1:1 mixture of Dulbecco’s Modified Eagle’s Medium (DMEM) and Ham’s F12 nutrient mixture. The Melan-A murine melanocyte cell line was generously supplied by Professor Dorothy C. Bennett (University of London, London, UK; ECACC, cat# 153599). Melan-A cells were grown in RPMI 1640 medium and 200 nM 12-O-tetradecanoylphorbol-13-acetate (TPA). The human melanoma cell line SK-MEL-1 was obtained directly from the LGC Standards/ATCC advanced cell catalog (ATCC, cat# HTB-67) and grown in Eagle’s Minimum Essential Medium (EMEM). All cell culture media were supplemented with 10% fetal bovine serum (Gibco BRL) and antibiotics consisting of 5% penicillin/streptomycin. Cultures were sustained at 37 °C in a humidified environment (95% relative humidity) with 5% CO_2_. Routine assays were performed to confirm that Mycoplasma spp. contamination was absent throughout the experimental period. Carnosic acid (CA) dissolved in DMSO (1 mg/mL) was diluted to a final concentration of 35 μM in phosphate-buffered saline (PBS). The final concentration of DMSO in culture media was ≤0.05% (*v*/*v*), a concentration verified to have no cytotoxic effect on the four cell lines in preliminary. For each well, 35 μL of the CA solution was added before experimental treatment. The administration of CA to the culture medium occurred 15 min before X-ray irradiation.

#### MTT and XTT Assays of Irradiated Cells

To analyze the radioprotective effects of the CA on PNT2, Melan-A and B16F10 cell lines, two MTT assay types were carried out as previously described [13,16]. MTT and XTT assays are colorimetric techniques used to assess cell survival. In our study, we used the MTT (3-(4,5-dimethylthiazole-2-yl)-2,5-diphenyltetrazolium bromide assay) for cell lines that adhere to the bottom of culture media and the XTT (2,3-bis(2-methoxy-4-nitro-5-sulfophenyl)-2H-tetrazolium-5-carboxanilide assay for cells that are kept as cells in suspension during the cell incubation period. Forty-eight hours post-incubation, cell proliferation was assessed following exposure to X-ray radiation. Briefly, PNT2 cells (3200 cells/well) and Melan-A and B16F10 cells (2500 cells/well) were seeded in 200 µL of growth medium and allowed to adhere to the well bottoms for 24 h. After 48 h of X-ray treatment, 50 µL of MTT solution (5 mg/mL) prepared in culture medium was added to each well, followed by incubation at 37 °C in a 5% CO_2_ atmosphere for 4 h. Subsequently, the culture medium containing unmetabolized MTT was removed.

For the XTT assay [13,16], SK-MEL-1 cells were seeded (2500 cells/well) and allowed to acclimatize for 48 h before treatment (20 Gy irradiation and/or addition of 35 μM CA). Following treatment, cells were incubated for either 24 or 48 h. The XTT assay was then performed by adding 50 μL of freshly prepared XTT solution to each well. This solution consisted of XTT (Sigma, 1 mg/mL) combined with PMS (5 mM, dissolved in PBS). The mixture was sterilized by passing through a 0.22-μm nylon membrane syringe filter and incubated at 37 °C in a humidified atmosphere containing 5% CO_2_ for 4 h.

After shaking the plates for 30 min at room temperature, the absorbance readings of the plates were read spectrophotometrically using a FLUOstar^®^ Omega spectrophotometer (BMG Labtech, Offenburg, Germany). Absorbance values at 570 nm and 690 nm were used for test and reference wavelengths, respectively. All experiments were performed in six independent replicates.

### 2.3. Annexin V

For apoptosis analysis by flow cytometry, the Alexa Fluor^®^ 488 Annexin V/Dead Cell Apoptosis Kit (catalog no. V13241) from Invitrogen™ (Thermo Fisher Scientific, Madrid, Spain) was used. This kit enables the detection of early apoptotic cells by identifying phosphatidylserine (PS) exposure on the outer leaflet of the plasma membrane and assesses membrane permeability, following the protocol described previously [14]. Stained cells were immediately analyzed using a FACSCalibur flow cytometer (Becton Dickinson), with fluorescence measured at 530 nm and 575 nm following excitation with a 488 nm laser. All experiments were performed in six independent replicates.

### 2.4. GSH Assay

The GSH/GSSG-Glo™ assay (Promega, Madison, MI, USA) was employed to measure and quantify total glutathione (GSH + GSSG), oxidized glutathione (GSSG), and the GSH/GSSG ratio in cells subjected to various experimental conditions 30 min following exposure to 20 Gy X-rays, as previously described [14]. The assay was performed according to the manufacturer’s instructions. Cell densities were assessed and normalized using the Bradford assay. The Bradford assay used bovine serum albumin (BSA, 0–100 μg/mL) as the standard, with absorbance measured at 595 nm. GSH and GSSG levels were normalized to total cellular protein content (μg protein) to account for differences in cell density across groups [17]. Cells were detached from the substrate by trypsinization, and fluorescence intensity was measured with the FLUOstar^®^ Omega system (BMG Labtech, Offenburg, Germany). All experiments included six independent replicates.

### 2.5. Cell Cycle

The cell lines were grown in 25 cm^2^ culture flasks with 5 mL of appropriate culture medium and incubated under standard conditions until adequate confluence was reached. The cells were then treated with fresh medium and CA was administered at a final concentration of 35 µM. Thirty minutes after compound administration, X-ray irradiation was performed. Forty-eight (48) hours following irradiation, the cells were suspended in culture medium and harvested by centrifuging at 200× *g* for 10 min at room temperature. After cell counting and adjustment to 1 × 10^6^ cells/200 µL in cold PBS, they were fixed with 2 mL of a 70% ethanol solution in PBS and kept for 30 min at 4 °C to achieve adequate cell permeability. Subsequently, the cells were centrifuged for 10 min at 4 °C at 1000 rpm and the supernatant was discarded. The pelleted cells were resuspended in 800 µL of cold PBS. For DNA staining, 50 µL of a propidium iodide (PI) solution at a final concentration of 400 µg/mL and 100 µL of RNase A (1 µg/mL) (Sigma-Aldrich, St. Louis, MO, USA) in PBS were added to each sample. PI was used at 50 μg/mL to avoid non-specific staining, as validated in preliminary experiments showing no overlap between G0/G1 and S phase peaks. The samples were incubated in the dark at 4 °C for 30 min to ensure correct intercalation of the PI with the cellular DNA. A FACSCalibur flow cytometer (Becton Dickinson, Franklin Lakes, NJ, USA) was used for data acquisition, using an excitation laser at 488 nm and an emission filter of 585/42 nm. DNA content histograms were generated and analyzed with ModFit LT 4.1 software (Verity Software House, Topsham, ME, USA), using the appropriate fitting model for determining cellular distribution across cell cycle phases (G0/G1, S, and G2/M). All experiments were performed according to the manufacturer’s instructions. All experiments were performed in six independent replicates.

### 2.6. Irradiation

An Andrex SMART 200E X-ray Generator (Yxlon International, Hamburg, Germany) was used, operating at 200 kV, 4.5 mA, 2.5 mm aluminum filtration, a focus to object distance (FOD) of 35 cm and a dose rate of 1.3 cGy/s. For the MTT, Annexin, cell cycle, and GSH assays, cell cultures grown in microplates were irradiated with a dose of 20 Gy. Radiation doses were continuously monitored inside the X-ray cabinet, with final dose validation conducted using thermoluminescent dosimeters. This study was approved by the Biosafety Committee in Experimentation of the University of Murcia (ID:472/2021), the Ethical Committee of the University of Murcia (CECA:510/2018; approved date: 14 December 2021), and the Government of the Autonomous Community of the Region of Murcia (Spain) (N° A13211208).

### 2.7. Statistical Analysis

In the cell survival assay, an analysis of variance (ANOVA) of repeated means was performed, and this was complemented by the least significant difference (LSD) test. In this case, we modified the previous formula [18] to adapt the percentage of Magnitude of Protection (MP) to cell survival:MP (%) = [(M_irradiated control_ − M_irradiated treated_/M_irradiated control_)] × 100 where M is the percentage of mortality with respect to non-irradiated control cells.

In the assessments of apoptosis and intracellular glutathione, analysis of variance (ANOVA) complemented with contrast analysis was applied to evaluate the relationships between variables. Additionally, quantitative means were compared using regression and linear correlation analyzes.

## 3. Results

Based on the experimental design outlined in Section 2, the following results were obtained:

### 3.1. Cytotoxicity Assay, MTT and XTT Assay

In all cell lines examined, treatment with carnosic acid (CCA) significantly altered cell survival compared to control cells (C) (*p* < 0.01), showing mild cytotoxicity after 48 h of incubation. Similarly, exposure to 20 Gy X-rays (Ci) led to a substantial reduction in cell survival by approximately 40% relative to controls (*p* < 0.001), demonstrating the cytotoxic effects of ionizing radiation on these cells. Administration of CA before irradiation (CiCA) produced different changes depending on the cell line tested. In PNT2 cells, the CiCA treatment significantly increased cell survival (*p* < 0.001) compared to irradiated cells (Ci), indicating a protective effect of CA against X-ray-induced cytotoxicity. A protection factor (MP) of 97 ± 1.1% was observed in these cells, reflecting CA’s radioprotective capacity by preventing 67% of radiation-induced cell death (*p* < 0.001), with survival levels comparable to non-irradiated controls (Figure 1). In contrast, the administration of carnosic acid (CA) to B16F10 cell cultures before irradiation (CiCA) resulted in a significant reduction in cell survival (*p* < 0.001), indicating an enhanced cytotoxic effect of X-rays compared to the irradiated control group (Ci). No protective effect of CA was observed in these cells. Rather, a decrease in cell survival by 32.1 ± 3.5% was noted, demonstrating the radiosensitizing effect of CA on B16F10 cells, effectively doubling cell death relative to irradiated cells (*p* < 0.001) (Figure 1).

In the Melan A and SK-MEL-1 cell lines, the administration of CA to cell cultures before irradiation (CiCA) does not show statistically significant differences with respect to the irradiated cells without CA (Ci). This demonstrates the absence of a cytotoxic effect induced by CA in these cells that would augment the cytotoxic effect induced by ionizing radiation, as determined in B16F10 cells. However, no radioprotective effect was observed, as determined in PNT2 cells.

### 3.2. Apoptosis

In the non-tumor cell lines, i.e., PNT2 and MELAN A, the administration of CA (CA) decreased apoptosis by ~60% in PNT2 cells and ~30% in MELAN A cells (both *p* < 0.001 versus respective controls); while in the tumor lines (SK-MEL-1 and B16F10) it produced an increase in apoptosis in both cell lines reaching statistical significance only in the SK-MEL-1 cells (*p* < 0.001) (Figure 2). Exposure to X-rays (Ci) was found to increase apoptosis in all cell lines tested (*p* < 0.001), the increase observed was nearly sixfold (6) in B16F10 cells, 2.3-fold in SK-MEL-1 cells, double (2-fold) in PNT2 cells, and rose by only 14% in MELAN A cells (Figure 2).

**Figure 2 cimb-47-00845-f002:**
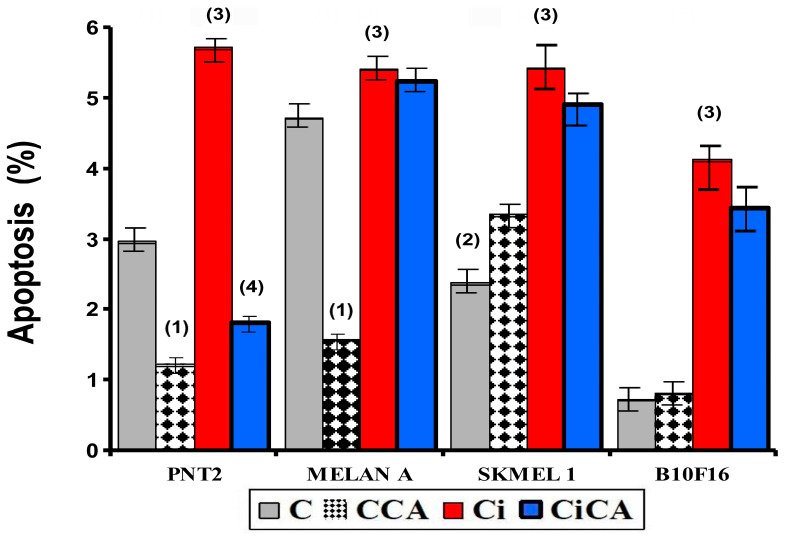
Percentage of PNT2, MELAN A, SK-MEL-1, and B16F10 cells in apoptosis after 48 h incubation periods (C, control; CCA, treated with carnosic acid; Ci, irradiated control; CiCA, cells were treated with CA before irradiation) ((1) *p* < 0.001 versus C; (2) *p* < 0.01 versus C; (3) *p* < 0.001 versus C; (4) *p* < 0.001 versus Ci). Data are mean ± SE of six independent experiments.

The administration of CA before exposure to X-rays (CiCA) produced a significant decrease in the percentage of apoptotic cells in PNT2 cells (*p* < 0.001), where apoptosis decreases by more than 300%, even below the percentage expressed by non-irradiated control cells (C). In the other cell lines, the reduction in the percentage of apoptotic cells was much lower: 3% in MELAN A, 10% in SK-MEL-1, and 17% in B16F10, which did not reach statistical significance (Figure 2 and Figure 3).

**Figure 3 cimb-47-00845-f003:**
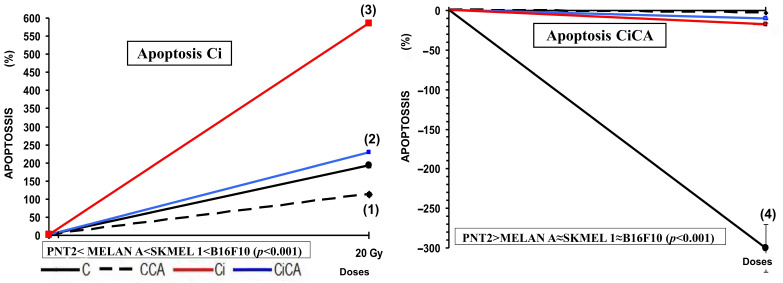
Percentage of PNT2, MELAN A, SK-MEL-1, and B16F10 cells in apoptosis 48 h after exposure to 20 Gy of X-rays in irradiated cells (Ci) and in irradiated cells treated with CA before the irradiation with X-rays (CiCA) ((1) *p* < 0.001 versus C; (2) *p* < 0.01 versus C; (3) *p* < 0.001 versus C irradiated; (4) *p* < 0.001 versus Ci). Data are mean ± SE of six independent experiments.

### 3.3. GSH Assay

The evaluation of total cellular GSH content showed statistically significant differences between the four cell lines studied. In the melanin-producing cells (MELAN A, SK-MEL-1, and B16F10) the amount of total glutathione determined in the non-irradiated cells (C) was significantly higher than that observed in PNT2 cells (*p* < 0.001). In B16F10 cells, the total glutathione concentration was more than twice what was measured in PNT2 cells (C) (*p* < 0.001) (Figure 4). The unexpected increase in total GSH in irradiated SK-MEL-1 and B16F10 cells may reflect adaptive upregulation of GSH synthesis (e.g., via γ-glutamylcysteine synthetase activation), a known radioresistance mechanism in melanoma [19,20].

In the PNT2 cells, the administration of CA (CCA) produced a significant decrease in the total glutathione (*p* < 0.001). Exposure to 20 Gy of X-rays (Ci) triggered a greater reduction in total glutathione (*p* < 0.001), which did not show any significant difference with the total glutathione observed in cells previously treated with CA before exposure to X-rays (CiCA) (Figure 4). These differences in total glutathione concentrations do not correlate with significant differences in the GSH/GSSG ratio, which may express a cellular capacity of the cells to eliminate damage caused by exposure to X-rays (Figure 5). In the melanin-producing cells (MELAN A, SK-MEL-1, and B16F10) the administration of CA (CCA) did not produce variations in total glutathione in B16F10 cells. However, it caused a decrease in glutathione content in MELAN A cells (*p* < 0.01) and an increase in SK-MEL-1 cells. Exposure to 20 Gy of X-rays (Ci) produced a reduction in total glutathione content (*p* < 0.001) in MELAN A cells while a significant increase in total glutathione was determined in SK-MEL-1 and B16F10 tumor cells (*p* < 0.001). However, the administration of CA before exposure to X-rays only produced a significant increase in total glutathione content in non-tumor cells and barely attained statistical significance in MELAN A cells (*p* < 0.001) (Figure 4). However, for these three cell lines (i.e., MELAN A, SK-MEL-1, and B16F10) exposure to X-rays produced a significant decrease in the GSH/GSSH ratio (Ci) (*p* < 0.01), which expresses damage induced by ionizing radiation. Furthermore, in two melanoma cell lines (i.e., SK-MEL-1 and B16F10), the administration of CA before irradiation (CiCA) did not show differences in the GSH/GSSH ratio with respect to irradiated cells (Ci), an expression of the absence of radioprotective effect of CA in these cells (Figure 5).

### 3.4. Cell Cycle

Exposure to X-rays (Ci) produces a significant increase in the percentage of cells in G2/M (*p* < 0.001) with a significant decrease in the G0/G1 phase (*p* < 0.001). This was accompanied by an increase in the S phase of the PNT2 line (*p* < 0.001) and a reduction in the S phase in the melanin-producing cells (MELAN A, SK-MEL-1, and B16F10) (Figure 6). Administration of CA to PNT2 cells had no significant effect on non-irradiated cells (CCA), while in the irradiated cells (CiCA), it reduced the intensity of the damaging effect elicited by X-rays (*p* < 0.001), which could be interpreted to mean a radioprotective effect. In addition, an increase in the population of cells in the G1 phase was observed, which could indicate apoptosis.

In the SK-MEL-1 cells, CA administration decreased the G2M cells (*p* < 0.001) by increasing the G0/G1 phase, which could be an expression of CA-induced cellular damage (CCA). In the irradiated cells, CA increased the cells in the G0/G1 phase (*p* < 0.001) while decreasing the G2/M phase, which could be considered a protective effect of CA (CiCA) (*p* < 0.001) (Figure 6). On the contrary, in the non-irradiated B16F10 cells, the administration of CA did not seem to have a significant effect (CCA), while in the irradiated cells (CiCA), it decreased the percentage of cells in the G0/G1 phase of the cell cycle (*p* < 0.01) with a concomitant increase in the G2/M phase (*p* < 0.01), so here, CA could be seen as a radiosensitizing agent.

In the MELAN A cells, the analysis was modeled manually, considering the aneuploid populations present (the model excluded debris and/or aggregates). Although the overall percentages of aneuploids and diploids were taken into account, the interpretation of the cell cycle results focused on the diploid population, as it is the most representative. In cells treated with the combination of irradiation and CA (CiCA), the apoptotic population was specifically modeled due to its relevance. In non-irradiated cells (CCA), a decrease in the G0/G1 phase was observed, accompanied by a significant reduction in the G2/M phase and an increase in S phase cells (*p* < 0.001), which could express a damaging effect of CA on MELAN A cells. In the irradiated cells (Ci), the reduction in G0/G1 phase cells was greater (*p* < 0.001) and there was a significant reduction in G2/M phase cells with an increase in S phase cells when CA is administered before irradiation (CiCA). This expresses a greater cellular damage than what was produced by ionizing radiation alone. These results indicate a complex and markedly different behavior from the PNT2 cells. CA alone (CCA) profoundly alters the cycle in MELAN A cells. In combination with irradiation, CA (CiCA) does not show a protective effect, but induces strong apoptosis and severely alters the cell cycle by decreasing the percentage of surviving cells (with massive accumulation in S phase (*p* < 0.001) and a block in G2/M phase (*p* < 0.01)). This suggests a potent cytotoxic and radiosensitizing effect of CA on the MELAN A cell line.

## 4. Discussion

CA has demonstrated significant radioprotective effects in various studies, attributed to its antioxidant properties—specifically, its ability to scavenge radiation-induced ROS [11,13,14,21]. Aruoma et al. [21] further confirmed that CA neutralizes hydroxyl and superoxide radicals, supporting its role in mitigating oxidative damage in non-tumoral cells. It has been reported to reduce the genotoxic and cytotoxic effects induced by ionizing radiation in various cell lines [9,13,14]. This radioprotective effect has been attributed to its ability to scavenge ionizing radiation-induced free radicals [9,13,14]. However, we previously reported a paradoxical radiosensitizing effect of carnosic acid in metastatic melanoma B16F10 cells and proposed a mechanistic action for this radiosensitizing effect that could provide a novel treatment strategy [14]. In this work, we examined the loss of the radioprotective capacity of CA in other cells and even the possible existence of a radiosensitizing effect in both normal and tumoral melanin-producing cells. Perhaps the analysis of these opposing effects could help clarify the mechanisms responsible for protecting healthy cells and simultaneously damaging neoplastic cells, thus achieving a desirable therapeutic strategy for cancer patients undergoing radiotherapy.

One of the parameters used to determine the cytotoxic effects induced by ionizing radiation and different chemical substances is cell growth inhibition. In radiobiology, this is also known as a cell survival assay [15,22]. In this study we used two assays to determine cell survival: the MTT assay for cell viability [9,10,16] and the Annexin V assay to determine apoptosis [11,14]. Several studies have reported that ionizing radiation induces cytotoxic effects in PNT2 cells [9,10,16]. However, studies on the effect of CA on cell survival in PNT2 cells exposed to IR are very scarce [13,14]. Our study confirms a significant decrease in cell survival after exposure to 20 Gy of X-rays and a significant increase in cell survival with prior administration of CA, indicating a high radioprotective capacity. This radioprotective capacity has been attributed to its ability to eliminate free radicals induced by ionizing radiation [9,10,16]. Similarly, some previous studies have reported the cytotoxic effect of ionizing radiation on B16F10 cells [9,10,16]. However, contrary to expectations, administering CA to B16F10 cells before irradiation resulted in a significant reduction in cell survival, indicating a radiosensitizing effect. This finding differs from the radioprotective effects of CA observed in other cell lines [9,10,14,16]. We have not sighted previous references on the effect of CA on cell survival in irradiated B16F10 cells [14]. However, various compounds have demonstrated antiproliferative effects, inhibiting cell growth and reducing metastatic invasion in vivo [23,24]. These effects are often enhanced when the compounds are administered before exposure [9,10]. Similarly, the combination of CA with other antitumor drugs presents an antiproliferative capacity in these B16F10 cells [25,26,27,28,29] showing that it inhibits adhesion and metastatic migration possibly due to the inhibition of the epithelial–mesenchymal transition (EMT) and the inactivation of AKT kinase [25,26,27,28,29]. We have not found previous references on the effect of ionizing radiation on cell survival in MELAN A and SK-MEL-1 cells. Our results show a reduction in cell survival caused by ionizing radiation similar to that determined in the other cell lines studied. We have also not found any references on the effect of CA on MELAN A and SK-MEL-1 cells. At the concentrations used in our study, CA produced a reduction in cell survival that is also similar in the four cell lines studied. However, in irradiated cells pretreated with CA, no radioprotective effect was observed in the two cancer cell lines, distinguishing them from normal prostate cells (PNT2). Although a reduction in cell survival was noted, it did not reach statistical significance. The extent of this loss of radioprotection on cell survival appears to vary in intensity, ranked from highest to lowest as follows: B16F10 < MELAN A = SK-MEL-1 < PNT2.

Cysteine and glutathione are the main intracellular antioxidants, capable of reducing oxidative stress by scavenging free radicals induced by ionizing radiation [30,31,32]. Numerous studies have established the relationship of cell resistance to treatment with chemotherapeutic substances and ionizing radiation with the amount of intracellular glutathione [12,23,24,30,31,32,33,34,35,36,37]. Our results show that B16F10, SK-MEL-1, and MELAN A cells have a much higher total glutathione concentration than PNT2 cells, confirming previously reported results [12,14]. These elevated concentrations of intracellular glutathione in normal melanocytes and melanoma cells have been used to explain the increased resistance of these cells to ionizing radiation and different chemotherapeutic agents [30,31,32,33,34,35,36,37]. In these cells, CA administration led to a reduction in the total glutathione (GSH) levels across all treatment groups (Ci, CA, and CiCA) compared to the control group (C), which may reflect the cytotoxic effects of CA on the cells. Nevertheless, a significant elevation in the GSH/GSSG ratio was observed in both irradiated and non-irradiated cells, potentially contributing to its radioprotective properties. This finding supports the hypothesis that CA acts synergistically or additively with intracellular endogenous glutathione to enhance cellular defense mechanisms [9,10,32,33]. References on the effect of CA administration on the total concentration of intracellular glutathione in MELAN A and SK-MEL-1 cells are not available. In previous studies, it was established that the administration of CA to B16F10 cells failed to modify the amount of total glutathione or the GSH/GSSG ratio [14]. However, administration of CA before X-ray exposure induced a significant reduction in the GSH/GSSG ratio, akin to that observed in irradiated cells. This decrease reflects a lower availability of reduced glutathione for neutralizing free radicals and correlates with an increased oxidative stress, indicating augmented cellular damage induced by ionizing radiation [12,14]. In the other two cell lines, i.e., MELAN A and SK-MEL-1, CA administration produced a similar response, leading to a significant decrease in the GSH/GSSG ratio, unlike the response observed in normal PNT2 cells. The degree of reduction in the GSH/GSSG ratio varies in intensity and can be ranked from lowest to highest as follows: B16F10 = MELAN A = SK-MEL-1 < PNT2. To date, we have not identified any studies that report the effect of CA on apoptosis in the cell lines used in this study. It has been reported that CA can increase apoptosis in different cell lines by inducing the expression of caspases 3, 8, and 9 and affecting the Akt/mTOR pathway [29,34,35]. In our study, CA decreased the percentage of apoptotic cells in non-tumor cell lines (PNT2 and MELAN A), did not modify the percentage of apoptosis in B16F10 and induced an increase in apoptotic cells in SK-MEL-1 tumor cells. However, prior administration of CA before X-ray exposure only reduced the percentage of apoptotic cells in PNT2 cells, with no significant differences observed in melanin-producing cell lines. The increase in cellular apoptosis varies in magnitude and can be ranked by intensity from highest to lowest as follows: PNT2 > B16F10 = MELAN A = SK-MEL-1.

In our study, exposure to X-rays produced characteristic changes in the cell cycle events such as an increase in the percentage of cells in G2/M as the most frequent alteration, both in human and murine cell lines, This is associated with the activation of p53 and p21, as well as alterations in the levels of Cyclin A and cyclin B1 [38,39,40,41,42]. Following radiation exposure, the proportion of cells in the G0/G1 phase either decreased or remained unchanged, and this was generally accompanied by a reduction in the S phase population [40,41,42]. Our results show these modifications in all irradiated cell lines, although we have determined differences in the S phase where the number of cells is similar in B16F10 cells, is reduced in MELAN A and SK-MEL-1 cells and only increases in PNT2 cells.

CA significantly reduced the G2/M phase of the cell cycle in several tumor cells, including Caco-2, HT-29, LoVo, and B16F10 [39,40,41]. This arrest prevents cells from entering mitosis, allowing for the activation of repair mechanisms or, in cases of irreparable damage, the induction of apoptosis. In some cellular contexts, CA induces G0/G1 arrest, associated with the overexpression of cyclin-dependent kinase inhibitors such as p21, which blocks progression to the S phase and DNA replication [38]. CA promotes the accumulation of cells in the sub-G1 phase, a marker of DNA fragmentation and apoptosis. This effect is dose-dependent and is associated with the activation of caspases and the cleavage of PARP [32,34].

We have not found any references reporting on the effect of CA administered before X-ray exposure in these cell lines. In our study on PNT2 cells, CA administration showed no effect on the irradiated cells. However, CA increased radiation-induced damage in B16F10 and MELAN A cells while reducing radiation-induced effects in SK-MEL-1 cells.

In exploring the possible explanation for the radiosensitizing effect of CA on melanoma cells, we found that the affected cells shared a common characteristic—the ability to produce melanin. In both normal and tumoral melanocytes, cysteine participates in the pheomelanin formation pathway in melanogenesis [29]. It has been reported that increased intracellular cysteine or glutathione concentrations activate the pheomelanin synthetic pathway to the detriment of other cellular activities [29,30,34]. In our study, the radiosensitizing effect mediated by the CA-induced decrease in the GSH/GSSG ratio may result from a combination of factors. Similarly to previous findings for caffeic and rosmarinic acids [10,12,14,21,34,43,44], CA could stimulate melanogenesis via the pheomelanin pathway, which consumes intracellular GSH. While CA may stimulate melanogenesis via the pheomelanin pathway (as observed for related phenolics like rosmarinic acid [12]), we did not measure pheomelanin levels or associated enzymes in this study—future work should validate this pathway. Consequently, the reduced glutathione pool diminishes, limiting its availability to neutralize ROS generated by ionizing radiation. Furthermore, in B16F10 cells, reduced superoxide dismutase activity has been reported, which may lead to decreased levels of intracellular reduced glutathione [28,34,35,45]. Additionally, CA may exert an inhibitory effect on enzymes such as glucose-6-phosphate dehydrogenase and glutathione reductase [45], as well as glutathione S-transferase—an effect observed with other diterpenes [45,46,47]. This inhibition could lower intracellular NADPH levels, thereby diminishing the cell’s ability to regenerate reduced glutathione from its oxidized form [12,14]. In addition, the composition of the culture medium, the concentration of fetal bovine serum, or pH may influence the cellular response to CA administration, modulating both cell viability and the magnitude of antioxidant, pro-oxidant, or radiosensitization effects observed after treatment [14,47]. CA could benefit from these conditions to enhance its effect on tumor cells and, like other alkylating agents and platinum-containing compounds, it would probably act better on cells located in acidic tumor beds [48,49]. Possibly the intensity of these effects could explain why the radiosensitizing effect was more pronounced in B16F10 cells. However, the loss of the radioprotective capacity of CA was observed in the three cell lines studied, whether tumoral or non-tumorous. Our results show that CA exhibits selective effects in melanoma models and studies are needed to validate these results in other different types of normal and tumor cells. Further studies are clearly needed to validate these findings and to gain a deeper understanding of the radiosensitizing potential of CA and its interaction with ionizing radiation-induced ROS. This should include assessments using various assays such as DPPH, iron-reducing antioxidant power, nitro blue tetrazolium (NBT) reduction, Western blot data for NRF2 nuclear translocation and antioxidant enzymes, and Comet assay or γ-H2AX staining for DNA damage repair, as well as the evaluation of detoxifying enzyme activities, including superoxide dismutase (SOD) and glutathione peroxidases (GPx).

In short, the synergy between cisplatin and other chemotherapeutics combined with ionizing radiation in melanoma is similar to the effect of CA combined with ionizing radiation [50]: both enhance tumor damage, modify apoptosis and arrest the cell cycle at critical phases, hindering DNA repair and promoting cell death. However, CA adds greater selectivity, since it protects normal cells while sensitizing melanin-producing cells, representing a relevant therapeutic advantage.

## 5. Conclusions

CA is a potent antioxidant compound that exhibits significant radioprotective properties. However, in melanin-producing cells (MELAN A and SK-MEL-1) it loses this radioprotective capacity or may even produce a paradoxical effect by becoming a radiosensitizing agent significantly reducing cell survival (B16F10). Our results show that CA exhibits selective effects in melanoma models; future studies should validate these findings in other types of normal and tumor cells. Understanding the mechanisms by which substances like CA exert their effects could facilitate the development of novel strategies that selectively protect healthy cells while mainly injuring neoplastic cells. This approach could represent a promising therapeutic strategy for cancer patients undergoing radiotherapy.

## Figures and Tables

**Figure 1 cimb-47-00845-f001:**
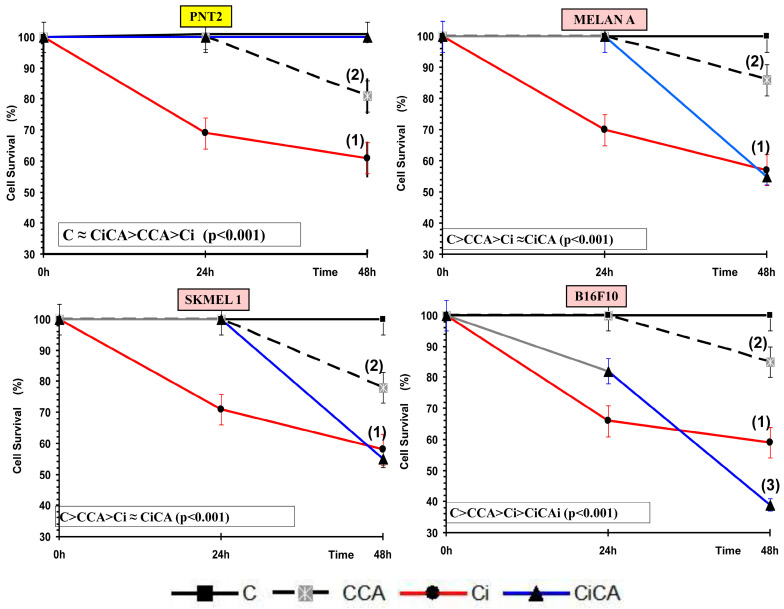
MTT and XTT assay: Cell survival curves for all studied cell lines (C, control; CA, treated with carnosic acid; Ci, irradiated control; CiCA, treated with CA before irradiation) ((1): *p* < 0.001 versus C; (2): *p* < 0.01 versus C; (3) *p* < 0.001 versus Ci). M_irradiated control = mortality of irradiated, untreated cells; M_irradiated treated = mortality of irradiated, CA-pretreated cells. Data are mean ± SE of six independent experiments.

**Figure 4 cimb-47-00845-f004:**
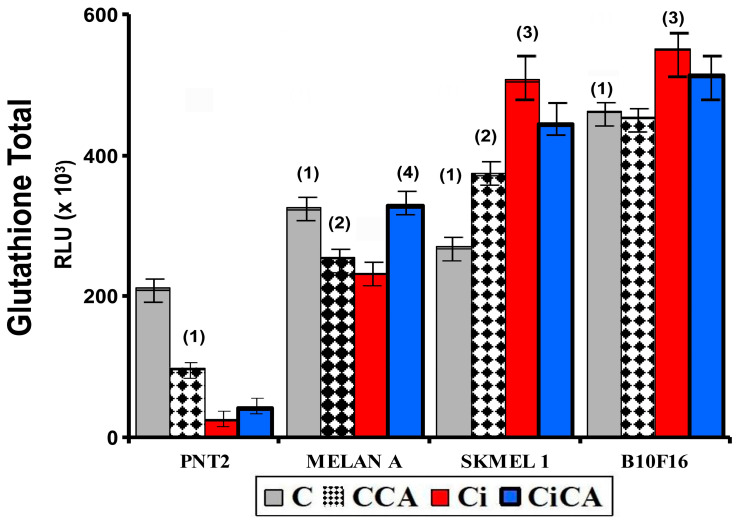
Total glutathione concentrations of the different groups studied (C, control; CCA, treated with carnosic acid; Ci, irradiated control; CiCA, cells were treated with CA before irradiation) ((1) *p* < 0.001 versus PNT2 control (C); (2) *p* < 0.001 versus PNT2 control (C); (3) *p* < 0.001 versus control cells, respectively (C); (4) *p* < 0.001 versus MELAN control irradiated (Ci)). Data are mean ± SE of six independent experiments.

**Figure 5 cimb-47-00845-f005:**
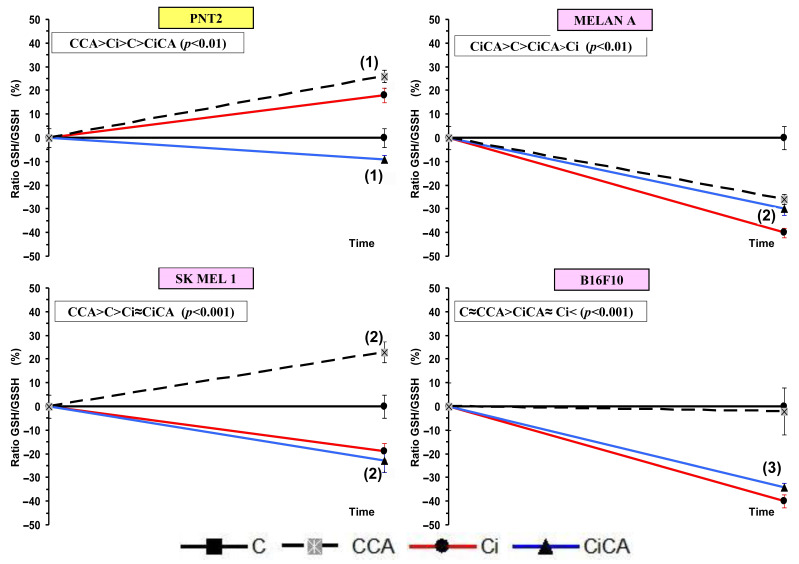
GSH/GSSG ratios of the different cell lines studied (C, control; CCA, treated with carnosic acid; Ci, irradiated control; CiCA, cells were treated with CA before irradiation) ((1) *p* < 0.05 versus C; (2) *p* < 0.01 versus C; (3) *p* < 0.001 versus C). Data are mean ± SE of six independent experiments.

**Figure 6 cimb-47-00845-f006:**
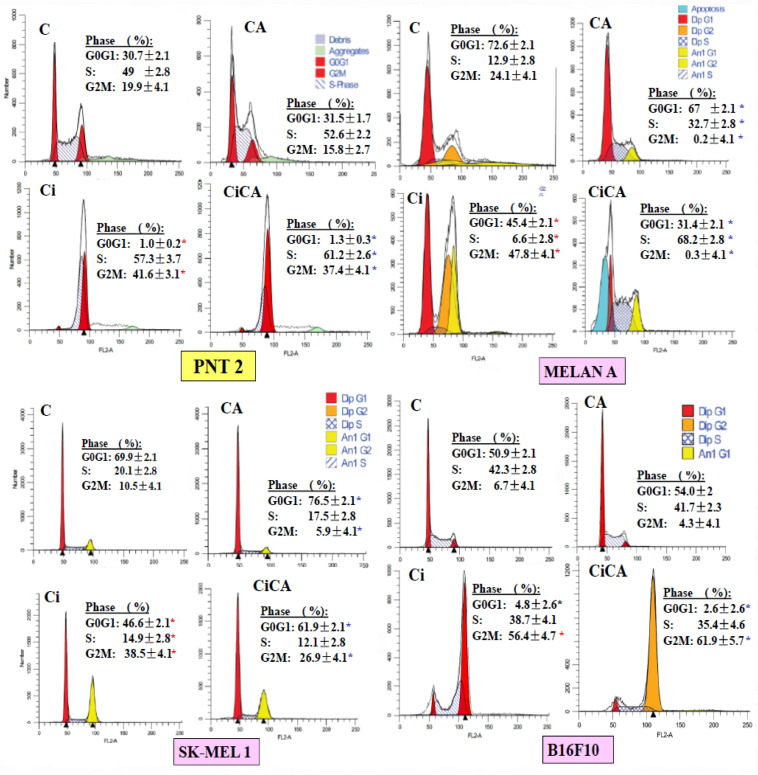
Cell cycle phases observed in the different cell lines studied (C, control; CA, treated with carnosic acid; Ci, irradiated control; CiCA, cells were treated with CA before irradiation) (*
*p* < 0.001 versus C; *
*p* < 0.001 versus Ci). Data are mean ± SE of six independent experiments.

## Data Availability

The original contributions presented in this study are included in the article. Further inquiries can be directed to the corresponding author.

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
