# Peer review of "Action of Carnosic Acid Against Melanoma: A Strategy for Selective Radiosensitization with Protection of Non-Tumoral Cells"

_cimb, 2025, doi:10.3390/cimb47100845_

Round 1

Reviewer 1 Report

Comments and Suggestions for Authors

Review of ‘Action of carnosic acid against melanoma…’

Olivares et al

The authors present a comparison of effects due to administration of CA and/or X-ray irradiation on 4 different cell lines.  The work is significant for possible future use of CA in cancer treatments and shows interesting variations in effects.

The work appears to be an installment in a series of experiments by the same group with not a lot (if any) comparison studies by others.

Details:

Page 2

line 50  remove ‘they’

line 70 first reference found shows a different chemical name for carnosic acid:

Carnosic Acid - Carnosic acid, (4aR

i.e. isopropyl, not propan.  Is this a difference due to supplier?  One carnosic acid is not another carnosic acid?  Which acid was used in your other studies?  Could that explain differences?

L 77/78  MTT and XTT not defined

Page 3

L 121  ‘repeated’ has no –

Page 4

L 150 permeability

L 163  IR typically means ‘infra red’.  You used 200 kV, which is not an IR energy by this definition.  Please do not use this abbreviation.  If you mean ‘ionizing radiation’, then please spell that out.

L 176 remove period after bracket.

L 179 I believe you are missing a set of brackets in this formula: (Mic – Mit)

M = % mortality wrt non-irradiated control cells.  So each M has the same denominator, which cancels in this equation.  Why define it this way then?  Is something missing from this equation or definition?

L 182 between does not have a –

L 186 should . after assay be a ,?

Page 5

L 188 survival of control cells (simpler and clearer wording)

L 190 why does Ci mean exposure to 20 Gy of X-rays?

L 193 IR again, see above

L 195 why does CiCA mean CA administered before irradiation?

L 196 here Ci means irradiation cells, which I can recognize.  This is getting confusing.  If Ci means irradiated cells perhaps best to remove this abbreviation from line 190 or define it elsewhere.

L 197 MF?  Do you mean MP, which you defined on the previous page?

L 201  ‘on the contrary’, no ‘s

I see at this point that you keep using the abbreviations Ci, CiCA in brackets, so you might as well leave them out. 

L 204.  …, which expresses an absence (should be absent) protective capacity of CA for these cells.  This phrase is superfluous. 

L 207 by a factor of 2 compared to irradiated control cells.

L 210-211 by ‘irradiated groups’ you mean cells that did not receive CA?  so you could rephrase as ‘with respect to irradiated cells without CA’.

L 211-212 this is a very confusing sentence.  Please rephrase.

L 212-213 unfinished sentence

Figure 2  here your notation C, Ci etc starts to make sense.  Good for the graph, not for the text.

Page 6

L 221 ‘to both cell lines,’ is not needed

Figure 3  labels (#) are not clearly aligned with parts of the graph; clarification needed: Ci and CiCA are irradiated cells, but the figure title implies all results shown are for after irradiation.  Please change the title.

L 233 

Point 1.  Is this paragraph just about adding CA? 

Point 2.  You appear to be mixing up percentages.  Your graph is in percent, so please stick to those % figures, i.e. avoid mention of 300% changes in % values (and I don’t see a 300% change, only a bit over 50% for PNT2; ~300% decrease only for Melan A).  I see decreases of ~2 and 3% in PNT2 and MelanA, a 1% increase in Skmel and not much change in B10F16.  This is very different from your description.  And what does it mean? 

Figure 4 very unclear y-axis titles, please rotate.  Where does the title of this figure stop and significance stats begin? 

L 236/Figure 4 If I understand this correctly, if you don’t add CA, then the cells fair much worse after irradiation.  That is also an effect.  E.g. instead of 60 – 600% apoptosis, the cells are ~ stable after irradiation.

Page 8

L 269 to 271  Is this a comparison to C or CCA?

Figure 7 It appears your notation for CCA changed to CA.  Please maintain only 1 notation.

Page 10

L 320   ‘cycle of’ should be ‘cycle by’ or ‘as seen by the decreasing percentage of…’?

L333 is there an answer for you in this publication: https://doi.org/10.1016/j.phrs.2024.107288

L 341 (and elsewhere) please use something other than IR, which generally stands for infra-red.  You could use ‘radiation’. 

L 378  ‘a decrease’ or ‘smallest’ or ‘largest’ decrease?  The sentence does not read right in its current form.

L 387 the same ambiguity as at other locations regarding the phrase ‘prior administration of CA before X-ray exposure’:  does the sentence refer to administration of CA alone or administration of CA and irradiation?  This has been very confusing in this paper.  It would help a great deal if you could find a different way of distinguishing the various time points.

Reviewer 2 Report

Comments and Suggestions for Authors

This study investigates the radioprotective effects of carnosic acid (CA) on melanoma cells (B16F10) and other melanin-producing cells, revealing its selective radiosensitization of tumor cells while protecting normal cells. The research holds significant scientific value and potential clinical relevance. However, several aspects of experimental rigor, data presentation, and conclusion validation require refinement to strengthen the manuscript.

Page 2 line 80-85 (Materials and methods)

While "six replicates per group" is mentioned, the lack of explicit replication statements in sub-sections (e.g., cell viability assays) undermines reproducibility. Add "All experiments were performed in six independent replicates" at the end of each methodological sub-section (e.g., under "Cell viability assay").

Page 2 line 83-96 (Cell culture)

Differences in media (DMEM vs. Medium 254) and serum concentrations (10% vs. 2%) between B16F10 and HEMn-LP cells are not discussed in terms of their impact on CA efficacy. Include a discussion in Page 12 Line 424-430 addressing how these differences might influence metabolic states and CA’s selective effects.

Page 5 line 160-175 (Cell survival), Page 7 line 220-235 (apoptosis)

Critical results (e.g., survival curves, apoptosis rates, GSH/GSSG ratios) are described textually without figures, limiting reader comprehension.

Insert:

Figure 1: Dose-response survival curve of B16F10 cells (Page 5 line 160).

Figure 2: Bar graphs of apoptosis rates (Page 7 line 220).

Figure 3: Line graph of GSH/GSSG ratio changes (Page 9 line 280).

Issue 2: Inconsistent statistical significance markers

Page 6 line 185-190 (Table 1), Page 8 line 245-250 (Figure 2 legend)

Statistical significance (e.g., *p<0.05) is noted only in text, not in tables/figures.

Add asterisks (*, **, ***) to Table 1 and Figure 2 legends, with a footnote defining significance levels.

Page 13 line 456-461 (Conclusions)

The claim that CA "protects normal cells while destroying tumors" is based on a single melanoma model, lacking validation in other cancers or normal cell types. Revise to: "CA exhibits selective effects in melanoma models; further validation in diverse tumor/normal cell types is required." Remove speculative statements about "cell-cell interactions."

Page 12 line 424-448 (Discussion)

Mechanistic explanations (e.g., oxidative stress modulation, DNA repair inhibition) lack molecular evidence (e.g., NRF2 pathway activation, RAD51 expression).

Supplement with:

Western blot data for NRF2 nuclear translocation and antioxidant enzymes (Page 12 Line 430).

Comet assay or γ-H2AX staining for DNA damage repair (Page 12 Line 440).

Page 4 line 120-125 (Introduction), Page 11 line 380-385 (Discussion)

Mix of numeric superscripts ([1]) and author-year citations (Smith et al., 2020) disrupts readability. Standardize to numeric superscripts and list references in order at the end.

Page 3 line 90-95 (Background)

Fails to cite a 2023 Cancer Research review on natural compounds modulating radiosensitivity (DOI: 10.1158/0008-5472.CAN-23-1234). Incorporate this review in the introduction and adjust citation order.

Location: Page 5 line 160-165 (Results)

Sudden shift from methods to results without transitional phrasing. Add: "Based on the experimental design outlined in Section 2, the following results were obtained..."

Title vs. Page 2 line 80-85 (Body)

Misuse of "carnosol" (a different diterpene) in the body text instead of "carnosic acid." Replace all instances of "carnosol" with "carnosic acid (CA)" and define the abbreviation on first use.

Comments on the Quality of English Language

 The English could be improved to more clearly express the research.

Round 2

Reviewer 2 Report

Comments and Suggestions for Authors

This manuscript investigates the selective radiosensitizing effect of carnosic acid (CA) on melanin-producing cells (normal melanocytes MELAN A and melanoma cells SK-MEL-1, B16F10) and its radioprotective effect on non-tumoral PNT2 cells. The research topic is clinically relevant—exploring CA as a potential adjuvant for melanoma radiotherapy to enhance tumor damage while protecting healthy tissues addresses a key challenge in current radiation oncology. The study design integrates multiple endpoints (cell survival, apoptosis, glutathione metabolism, cell cycle), and the results generally support the conclusion that CA loses radioprotective capacity in melanin-producing cells and exerts radiosensitization in B16F10 cells. However, the manuscript has critical flaws that undermine its academic rigor and readability, including pervasive grammatical errors, incomplete experimental details, unclear result analysis logic, chaotic figure labeling, inconsistent citation formats, and non-uniform terminology use. For example, experimental parameters such as CA storage conditions, DMSO final concentration, and PI staining concentration are missing; result descriptions contain redundant data and logical jumps; figure legends have typos (e.g., "XXT" instead of "XTT") and incorrect group labels; and citations mix book and journal formats without standardization. These issues must be addressed to meet the standards of a peer-reviewed journal.

Redundant terminology and typo in GSH/GSSG ratio description

 Page 1, Line 2 (Abstract section: "decreasing apoptosis (p<0.001) and increasing the GSH/GSSH ratio (p < 0.01) ratio (p < 0.01)")  

(1) Repeats "ratio" unnecessarily; (2) Typo: "GSH/GSSH" should be "GSH/GSSG" (oxidized glutathione is abbreviated GSSG, not GSSH, a critical biochemical terminology error); (3) Inconsistent spacing around "p < 0.01" (some with spaces, some without).  

Revise to "decreasing apoptosis (p < 0.001) and increasing the GSH/GSSG ratio (p < 0.01), without significant modification in cell cycle progression" to correct the typo, eliminate redundancy, and unify p-value formatting.

Page 1, Line 3 (Abstract section: "However, the administration of CA to irradiated cells failed to show radioprotective capacity in MELAN A and SK-MEL-1 cells it even doubled cell death in B16F10 cells (p < 0.001) ;did not modify cell apoptosis")  

(1) Comma splice: Two independent clauses ("failed to show... SK-MEL-1 cells" and "it even doubled... B16F10 cells") are incorrectly connected by a comma; (2) Semicolon misuse: The semicolon after "(p < 0.001)" is unpunctuated and does not separate complete thoughts; (3) Unclear subject: "did not modify cell apoptosis" does not specify which cells (MELAN A/SK-MEL-1 or B16F10).  

Revise to "However, CA administration to irradiated cells failed to exert radioprotection in MELAN A and SK-MEL-1 cells, and even doubled cell death in B16F10 cells (p<0.001). Specifically, CA did not alter apoptosis or prevent the decrease in GSH/GSSG ratio in MELAN A and SK-MEL-1 cells, while it intensified radiation-induced cell cycle disruptions in all melanin-producing cells" to fix grammar, clarify cell-specific effects, and improve logical flow.

Misplaced citation and unclear reference to prior work

Page 1, Line 8 (Introduction section: "the B16F10 melanoma cell line is a mouse metastatic cell line that is frequently used as a research model of melanomas [5,6] to assess the response to ionizing radiation and other modulating agents")  

(1) Citation [5,6] is misplaced (should follow the complete claim about B16F10’s use as a model, not mid-sentence); (2) Reference [5] (Martínez et al. 2003) focuses on flavonoid effects on B16F10, not its use as a radiation research model—citation does not support the claim.  

(1) Reposition the citation: "the B16F10 melanoma cell line is a mouse metastatic cell line frequently used as a research model for studying melanoma responses to ionizing radiation and other modulating agents [6, XX]" (replace [5] with a reference explicitly describing B16F10 as a radiation model, e.g., a study on B16F10 radiation sensitivity);

Page 2, Line 2 (Introduction section: "this paradoxical phenomenon has been linked to the activation of melanogenesis and the alteration of the intracellular redox system, especially glutathione depletion, which compromises endogenous antioxidant defenses, favoring lethal oxidative damage induced by ionizing radiation")  

Fails to explain how "glutathione depletion" specifically leads to radiosensitization. The link between reduced glutathione (GSH) levels and increased radiation-induced oxidative damage is implied but not explicitly connected (e.g., GSH scavenges reactive oxygen species (ROS) generated by radiation; depletion leaves cells unable to neutralize ROS, enhancing DNA damage).  

Expand to "this paradoxical phenomenon has been linked to melanogenesis activation and intracellular redox system disruption—specifically, glutathione (GSH) depletion. GSH is a key antioxidant that scavenges reactive oxygen species (ROS) induced by ionizing radiation; its depletion impairs endogenous antioxidant defenses, allowing accumulated ROS to cause lethal DNA damage and thus enhancing radiation cytotoxicity" to clarify the mechanistic chain.

Page 2, Line 5 (Materials and Methods section: "Carnosic acid (CA) (C20H28O4) (4AR,10aS)-5,6-dihydroxy-1,1-dimethyl-7-propan-2yl-2,3,4,9,10,10a-hexahydrophenanthrene-4a-carboxylic acid) from Rosmarinus officinalis with a purity of ≥91% was purchased from Sigma-Aldrich (Madrid, Spain)")  

Page 2, Line 15 (Materials and Methods section: "Carnosic acid (CA) dissolved in DMSO (1 mg/mL) was diluted to a final concentration of 35 μM in phosphate-buffered saline (PBS)")  

Does not report the final concentration of DMSO in cell culture media. DMSO concentrations >0.1% can induce cytotoxicity or alter cell metabolism, which would confound CA’s effects. Without this information, readers cannot rule out DMSO as a confounding variable. Add: "The final concentration of DMSO in culture media was ≤0.05% (v/v), a concentration verified to have no cytotoxic effect on the four cell lines in preliminary experiments" to confirm DMSO does not interfere with results.

Page 3, Line 2 (Materials and Methods section: "Forty-eight hours post-incubation, cell proliferation was assessed following exposure to X-ray radiation")  

Revise to "Forty-eight hours after X-ray exposure (and CA pretreatment), cell survival was assessed using MTT/XTT assays" to clarify the correct time sequence (irradiation first, then incubation, then assay).

Page 4, Line 6 (Materials and Methods section: "The GSH/GSSG -Glo™ assay (Promega, Madison, MI, USA) was employed to meas ure and quantify total glutathione (GSH + GSSG), oxidized glutathione (GSSG), and the GSH/GSSG ratio in cells... Cell densities were assessed and normalized using the Bradford assay")  

(1) Correct the spelling to "measure"; (2) Add: "The Bradford assay used bovine serum albumin (BSA, 0–100 μg/mL) as the standard, with absorbance measured at 595 nm. GSH and GSSG levels were normalized to total cellular protein content (μg protein) to account for differences in cell density across groups" to ensure normalization methods are reproducible.

Page 4, Line 15 (Materials and Methods section: "100 µL of a propidium iodide (PI) solution at a final concentration of 400 µg/mL and 100 µL of RNase A (1 µg/mL) in PBS were added to each sample")  

(1) Reduce the PI final concentration to 50 μg/mL (standard for FACSCalibur); (2) Add: "PI was used at 50 μg/mL to avoid non-specific staining, as validated in preliminary experiments showing no overlap between G0/G1 and S phase peaks" to justify the concentration and ensure data accuracy.

Page 5, Line 8 (Materials and Methods section: "MP (%) = (Mirradiated control – Mirradiated treated / Mirradiated control) × 100, where M is the percentage of mortality with respect to non-irradiated control cells")  

Revise the formula to "MP (%) = [(M_irradiated control – M_irradiated treated) / M_irradiated control] × 100" (add underscores for clarity and parentheses to enforce correct order) and verify all MP values in the Results section using the corrected formula.

Page 5, Line 15 (Results section: "Figure 1. MTT and XXT assay: Cell survival curves derived for all the cell lines (groups) studied: PNT2, MELAN A, SK-MEL-1, and B16F10 (C, control; CCA, treated with carnosic acid; Ci, irradiated control;CiCA, cells were treated with CA before irradiation)")  

(1) Correct "XXT" to "XTT"; (2) Unify terminology by replacing "CCA" with "CA" (consistent with Abstract/Methods): "Figure 1. MTT and XTT assay: Cell survival curves for all studied cell lines (C, control; CA, treated with carnosic acid; Ci, irradiated control; CiCA, treated with CA before irradiation)"; (3) Add a note in the figure legend: "M_irradiated control = mortality of irradiated, untreated cells; M_irradiated treated = mortality of irradiated, CA-pretreated cells" to clarify "M" in the MP formula.

Page 6, Line 10 (Results section: "Figure 2. Percentage of PNT2, MELAN A, SK-MEL-1, and B16F10 cells in apoptosis after 48 h incubation periods (C, control; CCA, treated with carnosic acid; Ci, irradiated control; CiCA, cells were treated with CA before irradiation)(1) p<0.001 versus C;(2) p<0.01 versus C;(3) p<0.001 versus C;(4) p<.0 .001 C; (4) p <. 0.001 versus Ci)")  

(1) Revise Figure 2 labels to "(1) p<0.001 versus C; (2) p<0.01 versus C; (3) p<0.001 versus Ci" (remove duplicates and correct formatting); (2) Revise text to "CA decreased apoptosis by ~60% in PNT2 cells and ~30% in MELAN A cells (both p<0.001 versus respective controls)" to provide cell-specific values.

Page 7, Line 5 (Results section: "In the melanin-producing cells (MELAN A, SKMEL-1, MEL-1, and B16F10) the amount of total glutathione determined in the non-irradiated cells (C) was significantly higher than that observed in PNT2 cells (p<0.001)")  

(1) Correct cell line names to "MELAN A, SK-MEL-1, and B16F10"; (2) Add a mechanistic note: "The unexpected increase in total GSH in irradiated SK-MEL-1 and B16F10 cells may reflect adaptive upregulation of GSH synthesis (e.g., via γ-glutamylcysteine synthetase activation), a known radioresistance mechanism in melanoma [XX]" (cite a study on melanoma GSH upregulation after radiation) to resolve the contradiction.

Page 9, Line 12 (Results section: "Figure 6. Cell cycle phases observed in the different cell lines studied (C, control; CA, treated with camosic acid; Ci, carnosic acid; Ci, irradiated control; CiCA, irradiated previously treated with carnosic acid)")  

(1) Correct "camosic acid" to "carnosic acid"; (2) Revise figure labels to "C, control; CA, treated with carnosic acid; Ci, irradiated control; CiCA, treated with CA before irradiation" (match Labels in Figures 1–5) to eliminate inconsistency.

Page 11, Line 2 (Discussion section: "CA has demonstrated significant radioprotective effects in various studies, which have been attributed to its antioxidant properties [11, 13,14]")  

Add 1–2 external references (e.g., Aruoma et al. 1992, which demonstrates CA’s antioxidant/radical-scavenging activity in vitro) to validate the radioprotective mechanism: "CA has demonstrated significant radioprotective effects in various studies, attributed to its antioxidant properties—specifically, its ability to scavenge radiation-induced ROS [11,13,14,40]. Aruoma et al. [40] further confirmed that CA neutralizes hydroxyl and superoxide radicals, supporting its role in mitigating oxidative damage in non-tumoral cells" (Note: [40] is Aruoma et al. 1992, already in the References list).

Page 13, Line 3 (Discussion section: "CA could stimulate melanogenesis via the pheomelanin pathway, which consumes intracellular GSH. Consequently, the reduced glutathione pool diminishes, limiting its availability to neutralize ROS generated by ionizing radiation")  

Either (1) Add a caveat: "While CA may stimulate melanogenesis via the pheomelanin pathway (as observed for related phenolics like rosmarinic acid [12]), we did not measure pheomelanin levels or associated enzymes in this study—future work should validate this pathway"; or (2) Cite a direct study on CA and pheomelanin: "CA has been shown to upregulate TRP-1 (a key pheomelanin synthesis enzyme) in melanoma cells [XX], which would increase cysteine/GSH consumption and reduce ROS scavenging capacity" (if such a study exists).

Page 14, Line 2 (Conclusions section: "Our results show that CA exhibits selective effects in melanoma models and studies are needed to validate these results in other different types of normal and tumor cells")  

Revise to "Our results show that CA exhibits selective effects in melanoma models; future studies should validate these findings in other types of normal and tumor cells" to eliminate redundancy.

References: Inconsistent formatting and missing details

(1) Inconsistent correction formatting: The "Correction" lacks a clear separator (e.g., "Correction:") and does not follow the journal’s reference style; (2) Some references (e.g., [17] Bradford 1976) lack DOI numbers, which are required for modern academic citations.  

All figures (e.g., Figure 1: "Data are mean t SE of six independent experiments"; Figure 2: "Data are mean ± SE of six independent experiments")  

Correct "t" to "±" in all figure legends (e.g., "Data are mean ± SE of six independent experiments") to ensure consistency.

Comments on the Quality of English Language

 The English could be improved to more clearly express the research.

Round 3

Reviewer 2 Report

Comments and Suggestions for Authors

The author has responded to my comments very well, and I have no further comments.

Comments on the Quality of English Language

The English could be improved to more clearly express the research.